# CHAROT: ROBUSTLY CONTROLLING CHAOTIC PDES WITH PARTIAL OBSERVATIONS

**Max Weissenbacher & Anastasia Borovykh**
Department of Mathematics & I-X
Imperial College London
London SW7 2AZ, United Kingdom
{m.weissenbacher, a.borovykh}@imperial.ac.uk

**Georgios Rigas**
Department of Aeronautics
Imperial College London
London SW7 2AZ, United Kingdom
g.rigas@imperial.ac.uk

## ABSTRACT

Control of chaotic partial differential equations is challenging but valuable, with far-reaching applications in energy systems, economics, fluid dynamics and many other domains. Realistic engineering applications often only admit partial observations of the state and the controller must learn to steer the system towards a desired state using incomplete information. We introduce CHAROT, an attention-based memory architecture designed to augment actor-critic reinforcement learning algorithms and improve their performance in controlling chaotic PDEs using only partial observations. We present numerical experiments for control of the Kuramoto-Sivashinsky equation in chaotic ($0.005 \leq \nu \leq 0.05$) and partially observable regimes. In the most chaotic regime considered, our method outperforms a memoryless controller by $150\%$ and an LSTM-augmented controller by $206\%$.

## 1  INTRODUCTION

Chaotic partial differential equations are ubiquitous in science and engineering, with applications in weather prediction, aerodynamics and the modeling of superconductors, to name a few. In this work, we are particularly inspired by the field of *closed-loop active flow control*, which involves dynamically manipulating the flow of fluids based on sensor measurements in order to improve the performance and efficiency of systems, e.g. actuating movable flaps attached to a vehicle to reduce its energy consumption (Brackston et al., 2016; Giannenas et al., 2022).

A fundamental challenge in flow control is the turbulent and hence chaotic nature of flows and the associated high-dimensional state space. In addition, for realistic flow control problems, the controller is only able to partially sense the flow field due to engineering constraints. For instance, pressure sensors might be located on a vehicle's body instead of in its wake. This partial observability leads to significant performance degradation compared to optimal sensor placement (Xia et al., 2023). There has been a recent surge in using deep reinforcement learning (RL) techniques applied to flow control (Vignon et al., 2023).

In reinforcement learning, the problem of partial observability is typically addressed by augmenting the controller with a 'memory', which is updated periodically using partial observations with the aim of inferring missing information about the current state. Methods can be roughly divided along two lines: memory based on *recurrent neural networks* such as LSTMs (Wijmans et al., 2023; Peng et al., 2018) and *attention-based methods* (Santoro et al., 2018; Parisotto et al., 2020; Pritzel et al., 2017). Memory-based algorithms are typically benchmarked on navigation problems and game-like environments (Morad et al., 2023; Oh et al., 2016). However, these benchmarks do not generalise in an obvious way to the control of chaotic PDEs. Moreover, RL applications in flow control predominantly assume near-optimal sensor configurations. When sub-optimal sensor placement and

therefore partial observability of the flow is considered, methods are typically limited to LSTMs and observation buffers.

In this work we consider the challenge of applying reinforcement learning to control the chaotic Kuramoto-Sivashinsky (KS) equation 1. The KS equation models a variety of unstable physical phenomena, such as chemical reaction-diffusion systems (Kuramoto & Tsuzuki, 1976), laminar flame front instabilities (Ashinsky, 1988) and hydrodynamic turbulence (Dankowicz et al., 1996) and is widely considered to be a proxy system for turbulence. We bridge the gap between the active flow control and reinforcement learning communities by introducing a novel attention-based memory architecture and studying it in the context of controlling a chaotic KS flow using only partial observations.

## 2 CONTROLLING THE KURAMOTO–SIVASHINKSY EQUATION

The Kuramoto–Sivashinksy (KS) equation is the fourth-order nonlinear partial differential equation:

$$\mathcal{L}(h) = h_t + h_{xx} + \nu h_{xxxx} + \frac{1}{2}(h_x)^2 = 0, \tag{1}$$

where $h : [0, \infty) \times [0, 2\pi] \to \mathbb{R}$ satisfies $h(t, 0) = h(t, 2\pi)$ for all $t \geq 0$, and $\nu > 0$ is a parameter. For $\nu < 1$, the KS equation exhibits complex spatio-temporal dynamics. As $\nu$ decreases, the trivial zero solution becomes unstable to periodic solutions, traveling wave solutions, oscillatory solutions and eventually chaotic solutions (Cvitanović et al., 2010; Kalogirou et al., 2015). Solutions are typically considered to be fully chaotic from about $\nu \leq 0.01$, for larger $\nu$ the periodic boundary conditions constrain the dynamics and 'simple' attractors exist (Wittenberg & Holmes, 1999).

To introduce control to the problem, let $A$ be the number of actuators and let $x_1, \ldots x_A \in [0, 2\pi]$ be equispaced points in the domain. We then define the controlled system

$$\mathcal{L}(h) = f(\vec{a}), \quad f(\vec{a}; x) = \sum_{i=1}^{A} a_i e^{-\frac{1}{2}\left(\frac{x-x_i}{\sigma}\right)^2}. \tag{2}$$

At each time step, the controller has access only to partial measurements of the solution $h(t, \cdot)$ from a limited number $S$ of point sensors in the domain. Let $\bar{x}_1, \ldots \bar{x}_S \in [0, 2\pi]$ denote $S$ equispaced points and let $\bar{h}(t) = (h(t, \bar{x}_1), \ldots, h(t, \bar{x}_S))$ be the vector consisting of $h(t, \cdot)$ evaluated at the sensor locations.

We aim to steer the system towards the trivial zero solution $h = 0$, starting from initial data $h_0$ which are a perturbation of the zero solution. For this, let us define the reward signal $r(t) = -\|h(t, \cdot)\|_{L^2}$. We remark that for each $\nu < 1$, the equation possesses finitely many steady-state solutions, of which the zero solution is the most unstable with $\lfloor 2\nu^{-\frac{1}{2}} \rfloor$ unstable real modes. In line with this, the Kaplan–Yorke dimension (a measure of the fractal dimension of the chaotic attractor) is conjectured to grow linearly with $\nu^{-\frac{1}{2}}$, a claim supported by numerous numerical studies, e.g. (Collet et al., 1993b; Edson et al., 2019). Figure 1 provides an overview of the RL feedback loop.

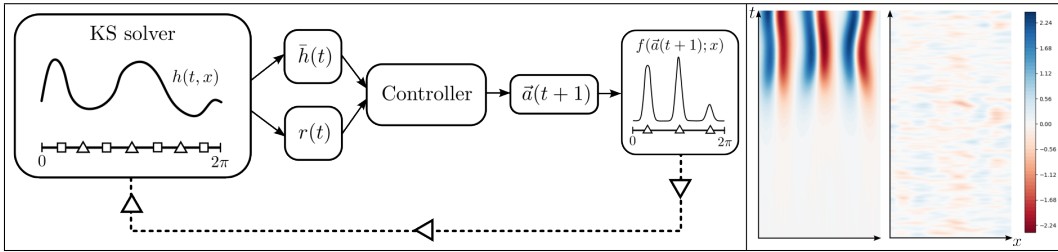

Figure 1: *Left*: Overview of the RL training loop. A numerical solver takes the mixture weights $\vec{a}(t)$ as input and steps equation 2 forward in time. The partial observation $\bar{h}(t)$ and reward $r(t)$ are fed into the RL controller, which computes a new mixture weight $\vec{a}(t+1)$ for the next time step. *Right*: Uncontrolled solution (left) and solution controlled with CHAROT (right) for $\nu = 0.05$.

Our setup closely mirrors typical practical engineering applications: As an example, consider a turbulent flow past a bluff body (such as a car), which is equipped with blowing and suction jets at

the rear. The aim is to actuate the jets in order to reduce turbulence or steer the wake. In a realistic application, sensors might only be located on the bluff body itself. This resembles our problem setting: a highly chaotic flow which can only be sensed at a limited number of points in the flow domain and which must be steered towards an unstable (laminar) flow state.

## 3 METHODS

To control the KS system, we use an actor-critic RL controller and introduce three variants of its actor and critic architecture: the original controller (no memory augmentation), the controller augmented with an LSTM and the controller augmented with our novel CHAos-RObust-Transformers (CHAROT). Figure 2 provides an overview of the agent architectures. The underlying controller used in this study is TQC, a maximum-entropy actor-critic algorithm (Kuznetsov et al., 2020), although we consider the particular controller architecture to be a black-box which may be replaced with other choices, e.g. with SAC (Haarnoja et al., 2018) or A3C (Mnih et al., 2016).

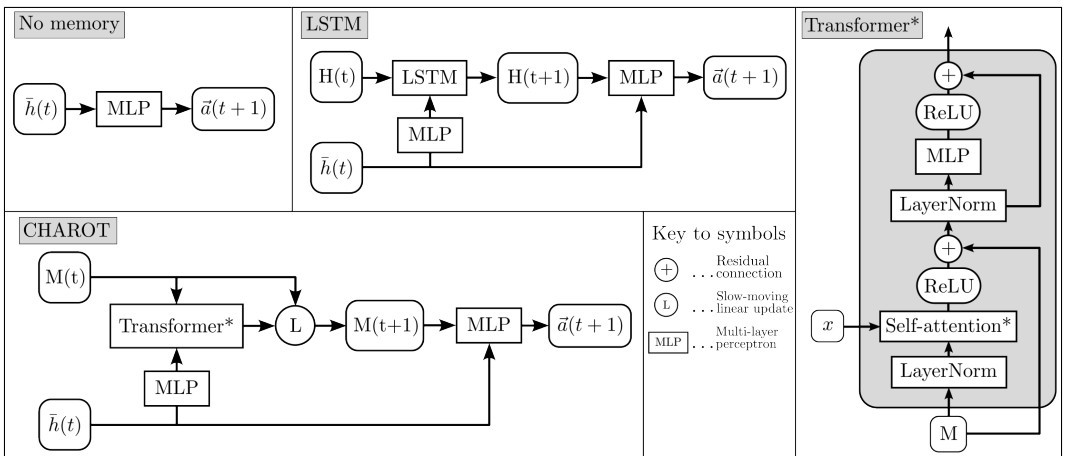

Figure 2: Schematic representation of the three controller variants considered. We only depict the actor part of the controller. At each time step, the actor receives the partial observation $\bar{h}(t)$ as input and outputs the control $\vec{a}(t + 1)$ for the next time step. *Top left*: the original actor architecture. *Top middle*: The controller augmented with an LSTM, whose hidden state $H(t)$ is updated each time step. *Bottom left*: Our novel architecture. The memory $M(t)$ and observation $\bar{h}(t)$ are mapped through a modified Transformer (denoted as Transformer*) and a slow-moving update to compute the updated memory $M(t + 1)$. *Right*: The modified Transformer component of CHAROT.

### 3.1 ATTENTION-BASED MEMORY

Our novel memory augmentation, CHAROT, introduces a time-dependent external memory $M(t)$ to the actor and critic of the controller. At its heart, the architecture of CHAROT consists of a modified Transformer (Vaswani et al., 2017) with a slow-moving update, see Figure 2. We limit ourselves to describing the modified self-attention mechanism and the slow-moving update, denoted respectively as Self-attention* and $L$ in Figure 2.

The memory $M(t)$ is a $n \times s$ matrix, where $n$ is the number of memories and $s$ is the size of each memory. At the start of each control episode, entries of the memory matrix are independently sampled from a unit normal distribution. At time $t$, the self-attention layer then receives as input the normalised memory $M = \text{LayerNorm}(M(t))$ from the previous time step and the pre-processed observation $x = \text{MLP}(\bar{h}(t))$. We assume that the MLP has final dimension $s$, so that $x \in \mathbb{R}^s$. We append $x$ as the last row to $M(t)$, forming the $(n + 1) \times s$ matrix $\tilde{M}(t)$. Then we compute:

$$Q(t) = M(t)W_q, \quad K(t) = \tilde{M}(t)W_k, \quad V(t) = \tilde{M}(t)W_v, \tag{3}$$

$$S(t) = \text{softmax}\left(\frac{Q(t)\,K(t)^T}{\sqrt{s}}\right), \quad O(t) = \tanh\left(S(t)\,V(t)\right), \tag{4}$$

where $W_q, W_k, W_v$ are learnable $s \times s$ matrices. The $n \times s$ matrix $O(t)$ is the output of the modified self-attention layer and is mapped through residual connections, layer normalisation and MLPs as indicated in Figure 2. We note that these subsequent layers act per-memory.

Let us denote the final output of the modified Transformer layer by $\bar{M}(t+1)$. In order to compute the updated memory $M(t+1)$, we perform a learnable slow-moving memory-wise update:

$$M(t+1) = \bar{M}(t+1)\,W_1 + M(t)\,W_2, \tag{5}$$

where $W_i = \sigma(\tilde{W}_i)$ are square $s \times s$ matrices with learnable parameters $\tilde{W}_i$ for $i \in \{1, 2\}$ and $\sigma$ is the Sigmoid function. The weights are initialised such that $W_i = \frac{1}{2} I_s$ for $i \in \{1, 2\}$, where $I_s$ is the $s$-dimensional identity matrix. The source code is available online[1].

## 4  EXPERIMENTS

Here we present the results of numerical experiments of stabilising the KS equation towards the zero solution. Initial data $h_0$ are chosen in a neighbourhood of the zero solution by setting $h_0$ to be white noise with amplitude $1 \times 10^{-2}$. Our experiments are characterised by two key factors:

**Fully and 'weakly' chaotic regimes.** Controllers are trained for $\nu$ in the range $[0.005, 0.05]$. Therefore, we study both the fully chaotic regime ($\nu \leq 0.01$) and the 'weakly' chaotic regime ($\nu \geq 0.01$), where 'simple' attractors exist, see Section 2. The number of unstable modes of the zero solution varies between 28 (for $\nu = 0.005$) and 8 (for $\nu = 0.05$). The maximal Kaplan-Yorke dimension considered is 20 for $\nu = 0.005$. This is contrast to previous studies applying RL to the KS equation, which have mostly focused on the weakly chaotic regime, see Appendix C.

**Partial observability.** For each choice of $\nu$ and $A$, there exists a critical number of sensors $S_c(\nu, A)$ such that the un-modified controller can robustly stabilise the system with $S \geq S_c$ sensors, and will fail to stabilise with $S < S_c$ sensors. This is backed up by additional experiments presented in Appendix A. Therefore, we consider the system to be fully observable when $S \geq S_c$, and partially observable otherwise. Our choice of $S = 10$ sensors and $A = 9$ actuators was made such that the system is fully observable for $\nu = 0.05$, and partially observable for $\nu \leq 0.04$.

We train for a total of $5 \times 10^6$ solver steps. We periodically evaluate controller performance during training; to evaluate at training time $T$ we compute a controlled solution $h_c(t, x)$ using the current model and compute the energy $E(T) = \|h_c\|_{L_t^1 L_x^2}$. To evaluate the performance of a given training run, we then compute $\mathcal{L} = \max_{T \geq 2 \times 10^6} E(T)$. This metric emphasises robustness and speed of convergence. The total number of trainable parameters is approximately $3 \times 10^6$ for all three controllers, see Appendix B.4. Figure 3 summarises the results of our numerical experiments.

In summary, we find that CHAROT outperforms the un-modified controller under partial observability ($\nu \leq 0.04$), with the performance gap generally increasing as the system becomes more chaotic. For the lowest $\nu$ value, $\nu = 0.005$, CHAROT's median performance surpasses the un-modified controller by $150\%$ and the LSTM by $206\%$. Augmenting with an LSTM underperforms compared to the unmodified, memoryless controller and shows high sensitivity to hyperparameter choices.

**Why does CHAROT perform better?** In the control of chaotic systems, small changes of the controller output may incur large changes in dynamics and the resulting observations. These small errors may accumulate quickly in the hidden state of an LSTM and prevent it from extracting a relevant latent state. The attention mechanism used to update the memory of CHAROT is more robust to small changes in observations, due to the averaging over past memories and new observations in the Transformer block. We therefore conclude that attention mechanisms such as Transformers are better suited to the control of chaotic systems than recurrent architectures such as LSTMs (hence CHAROT = Chaos-Robust-Transformers). This is unexpected as RNNs are typically the architecture of choice for enhancing controllers with memory.

---

[1]https://github.com/maxweissenbacher/charot

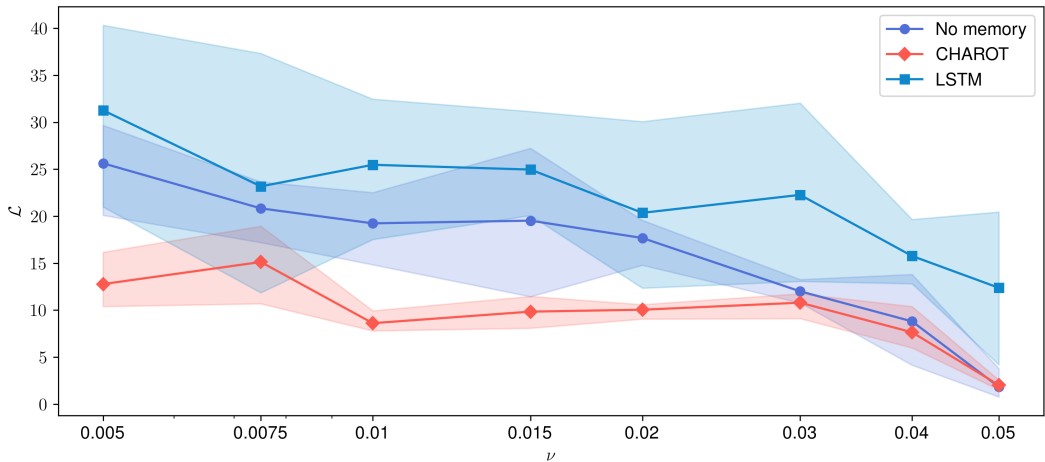

Figure 3: Medians of $\mathcal{L} = \max_{T \geq 2 \times 10^6} E(T)$ over $5$ repeated training runs. The filled areas indicate minimum and maximum values. We used CHAROT with $n = 10$ memories of size $s = 20$.

ACKNOWLEDGMENTS

The authors would like to thank Joseph Lee and Eleanor Broadway for setting up the Kubernetes cluster and providing invaluable trouble shooting support. The authors would further like to thank Elise Özalp for a fruitful discussion on the use of LSTMs for the Kuramoto–Sivashinksy equation. This research has been funded through the UKRI AI for Net Zero grant "Real-time digital optimisation and decision making for energy and transport systems" (EP/Y005619/1).

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

## A  FURTHER RESULTS

### A.1  SENSOR ABLATION AND PARTIAL OBSERVABILITY

Here we study the dependence of performance on the number of sensors $S$ available to the controller. Intuitively, reducing the number of sensors (while leaving all other system paramters unchanged) should make the problem more difficult. We present results here that show that for each choice of $\nu$ and number of Gaussian actuators $A$, the performance of the un-modified controller decreases as the number of sensors decreases. This leads us to define a critical number of sensors $S_c$ as the minimal number of sensors for which unmodified TQC stabilises the system.

We note that $S_c$ depends on the parameter $\nu$ of the KS equation, the number of Gaussian actuators $A$ and their width $\sigma$, a notion of what 'stabilising the system' means precisely, as well as the baseline

controller used. (For instance, replacing TQC with SAC will lead to slightly different results.) Thus $S_c$ denotes the point at which a particular (memoryless) algorithm fails to stabilise the system.

Figure 4 contains the results of repeated training runs of the un-modified TQC controller, using different numbers of sensors and various choices of $\nu$. As expected, the performance degrades as the number of sensors reduces. While we refrain from computing $S_c$ explicitly here, we note that for $S = 10$ sensors, the unmodified TQC algorithm stabilises the system successfully for $\nu = 0.05$, but fails to stabilise it for $\nu \leq 0.04$, so that certainly $S_c(\nu) > 10$ for $\nu \leq 0.04$. We note that other than varying the number of sensors, we used the same parameters as in the main body of the text to create Figure 4.

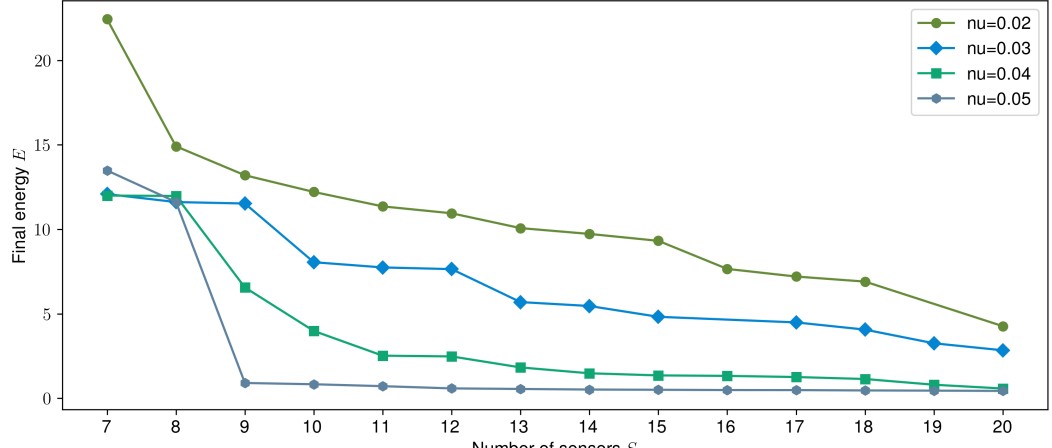

Figure 4: The final energy $E$ as a function of varying numbers of sensors for the un-modified TQC controller. For $S = 10$ sensors, performance is poor for $\nu \leq 0.04$, hence we refer to this setting as 'partially observable'. On the other hand, the system is stabilised by the unmodified controller using $S \geq 9$ sensors for $\nu = 0.05$, hence we refer to this setting as 'fully observable'.

## A.2 THE EFFECT OF SYSTEM PARAMETERS ON PROBLEM DIFFICULTY

While the dynamics of the (uncontrolled) KS equation are completely determined by a choice of $\nu$ and initial condition $h_0$, the control problem depends on a number of further parameters. In order to fully specify the control problem, one must choose the number of actuators $A$, the number of sensors $S$, the positions of the actuators and sensors in the domain, the width (or standard deviation) $\sigma$ of each Gaussian actuator, as well as the frequency at which the controller interacts with the system ('frame skip'). Intuitively, the difficulty of controlling the system depends on these parameters (e.g., allowing less frequent interactions or a lower number of sensors is likely to make the problem more difficult). In practice, we have found that the performance of both un-augmented TQC and the controller augmented with CHAROT depends in a complex way on these system parameters.

## A.3 ARCHITECTURAL CHOICES

We explored a variety of architectural choices throughout the development process. While we do not have quantitative data to present a precise comparison of the effect on performance of all the architectural choices we considered, we include here some qualitative remarks.

**Separate memories.** We found using separate memories for actor and critic necessary to achieve convergence. Sharing the memory between the actor and critic network led to a complete breakdown of convergence. We tested sharing the the memory between actor and critic, and allowed both the actor and critic loss to update the memory component's parameters. We did not explore using a stop gradient on either the actor or critic loss to prevent the memory from being updated by both actor and critic.

**Two memories are better than one.**  We found that using a memory only for the actor (and not the critic) decreased performance by a large margin. The authors have not tested using a memory only for the critic and not the actor. We leave a more precise investigation of how the actor and critic networks use the memory to future work.

**Size of the memory matrix.**  In our numerical experiments, we used a memory matrix $M$ of size $n \times s$, where $n = 10$ and $s = 20$. We found this to work well across several values of $\nu$. However, we point out here that there is potential for tuning the size of the memory matrix depending on the degree of chaos in the system (i.e. the choice of $\nu$). We leave this for future investigations.

**Reordered Transformers.**  Parisotto et al. (2020) report that a reordering of the Transformer layers, known as "identity reordered Transformer" performs significantly better on the DMLab-30 benchmark (Beattie et al., 2016) than the standard ordering of Transformer layers. We use this Transformer architecture in our experiments. However, we did not observe a significant difference in performance between the 'original' Transformer (Vaswani et al., 2017) and the reordered Transformer from (Parisotto et al., 2020). Just like for regular Transformers, one could make use of several Transformer layers and attention heads. We have found that using only one Transformer layer and one attention head performed better than several.

**Design of the reward function.**  Commonly in the control of PDEs, the reward function includes a term which penalises the size of the actuations. This essentially amounts to limiting or minimising the total amount of energy the controller expends and often has a regularising effect on training performance. Our experiments have shown that adding a term of the form $\alpha \|\vec{a}(t)\|$ or of the form $\alpha \|\partial_t \vec{a}(t)\|$, for some hyperparameter $\alpha > 0$ to the reward function $r(h) = -\|h(t, \cdot)\|_{L^2}$ did not have a positive impact on performance. We leave this subject for future investigations.

## B  IMPLEMENTATION DETAILS

### B.1  DIFFERENCES BETWEEN ACTOR AND CRITIC MEMORY

Our architecture is designed to augment both the actor and the critic components of the underlying RL controller. In Section 3 above we described in some detail the structure of the augmentation for the actor. We note here that the augmentation for the critic is virtually identical, the only difference being that in addition to receiving the latest partial observation $\bar{h}(t)$ as input, the critic also receives the action $\vec{a}(t + 1)$ computed by the actor in this timestep as an additional input. These two inputs are then concatenated and mapped through an MLP together, $x = \text{MLP}(\bar{h}(t), \vec{a}(t + 1)$. This vector $x \in \mathbb{R}^s$ is then used in an identical manner to the actor critic to compute the updated memory $M(t + 1)$. In addition, the final output of the critic is computed by feeding $\bar{h}(t)$, $\vec{a}(t + 1)$ and the updated memory $M(t + 1)$ into the final MLP layer.

### B.2  SIMULATION ENVIRONMENT

We solve the KS equation using a third-order Runge–Kutta scheme. The numerical solver was adapted from the pyks solver (Whitaker, 2015) and modified to work with PyTorch. Note that we are using the integral form of the KS equation, its derivative form (obtained by taking an $x$-derivative of $h$) is also commonly studied. We used a timestep of $dt = 0.005$ to propagate the equation forward in time and chose $N = 64$ Fourier nodes, which was sufficient to resolve the system for $0.005 \leq \nu \leq 0.05$.

To the reader interested in well-posedness, we note that the Cauchy problem for the KS equation is well-posed (Tadmor, 1986) for periodic initial data $h(0, x) = h_0(x)$. Moreover, the KS equation possesses a smoothing property similar to the heat equation: periodic initial data $h_0 \in L^2([0, 2\pi])$ give rise to solutions which are real analytic for all positive times (Collet et al., 1993a). By choosing white noise initial data (on the discretised domain) we are therefore ensuring existence, uniqueness and smoothness of the solution for all times.

The RL controller interacts with the KS equation every $F$ steps, where $F$ is an integer. We refer to $F$ as the 'frame skip', a term commonly used in the RL literature. In between interactions with the RL controller, the action which is actually executed is a linear interpolation from the last action used

to the new action chosen by the controller. Therefore, the actions are being executed in a 'slow-moving' or smooth manner and it takes $F$ steps until the action chosen by the controller is fully executed. This mimics the way in which real-world engineering systems function, where physical actuators exhibit inertia and cannot be changed or moved suddenly to prevent damage.

The actor component of the RL controller (in all three variants) is probabilistic in the following sense: During training time, instead of computing a value for each mixture weight $a_i$ directly, the actor network outputs a mean $m_i$ and standard deviation $\sigma_i$. We then sample $\tilde{a}_i \sim \mathcal{N}(m_i, \sigma_i)$ and finally compute $a_i = \tanh(\tilde{a}_i)$. This serves to enhance the controller's exploration of the action space during train time. During evaluation time, the controller becomes deterministic and we choose $a_i = m_i$.

## B.3 COMPUTATIONAL ENVIRONMENT

The numerical experiments were carried out on the Edinburgh EIDF cluster, using Nvidia A100 GPUs and on the Imperial College high performance computing service, using Nvidia Quadro rTX 6000s. Our code is implemented in the torchrl package (Bou et al., 2023) and is available online. For a frameskip parameter between 10 and 20, training takes between 4 and 6 hours for $5 \times 10^6$ time steps in total on a single GPU.

## B.4 TRAINING DETAILS AND HYPERPARAMETERS

Training is carried out in an episodic manner. After each episode, the parameters of the model are updated with $G$ gradient updates using the ADAM optimiser, where $G$ is the episode length divided by the frameskip. Before training commences, we take $2.5 \times 10^5$ random actions without updating model parameters. This serves to increase the controller's exploration of the action space. Each model is trained for a total number of $5 \times 10^6$ time steps. We summarise the most important hyperparameters in Table 1. Note that all hyperparameters concerning the TQC algorithm are exactly as in the original publication (Kuznetsov et al., 2020), so that we do not include these in Table 1.

In the table, we refer to 'pre-processing MLPs'. These refer to the MLP located before the LSTM respectively the modified Transformer, see Figure 2. We note carefully that in each of the three variants (unmodified, LSTM, CHAROT), there is a final MLP layer which computes the final output of the actor respectively critic. This final MLP layer is identical in all three cases, and is in fact identical with the MLP layers used in the un-modified TQC algorithm (Kuznetsov et al., 2020). Therefore, we refrain from providing details on the final MLP layers. All activation functions used are ReLU. We summarise the total number of trainable parameters for each model used in Table 2.

Table 1: Hyperparameters

| General hyperparameters | |
|---|---|
| Frameskip | 20 steps |
| Width $\sigma$ of Gaussian actuators | 0.3 |
| Episode length | $10^4$ steps |
| Batch size | 2048 |
| Gradient updates (per episode) | $5 \times 10^3$ |
| Total training time steps | $5 \times 10^6$ steps |
| Frequency of model evaluation | $2.5 \times 10^5$ steps |
| **CHAROT parameters** | |
| Number of memories $n$ | 10 |
| Memory size $s$ | 20 |
| Weight decay of ADAM optimiser | 0 |
| Preprocessing MLP hidden size | 128 (single hidden layer) |
| Transformer MLP hidden sizes | $[20, 20]$ (two hidden layers) |
| **LSTM parameters** | |
| LSTM hidden size | 256 |
| Weight decay of ADAM optimiser | $1 \times 10^{-6}$ |
| Preprocessing MLP hidden size | 128 (single hidden layer) |

Table 2: Total number of trainable model parameters of TQC augmentations

| No memory | 2 815 119 parameters |
|-----------|----------------------|
| LSTM | 2 964 495 parameters |
| CHAROT | 3 398 167 parameters |

## C  RELATED WORK

Bucci et al. (2019) control the KS equation towards non-trivial steady state solutions using the Deep Deterministic Policy Gradient (DDPG) algorithm (Lillicrap et al., 2015). The authors study the KS equation on a domain of length $L = 22$, which corresponds in our setting to a fixed choice of $\nu = 0.08156$ (rounded). For this value of $\nu$, the dynamics of the system are confined to oscillations between a set of six steady state and travelling wave solutions and can therefore not be considered to be truly chaotic.

Gomes et al. (2017) apply optimal control to a generalised KS equation including an electric field and dispersion. Their method proceeds by considering the Galerkin decomposition of the solution and proving that there exists a linear combination of a finite number of Galerkin coefficients which stabilizes the system. They demonstrate that their linear feedback control can stabilise the KS equation to arbitrary steady state, travelling wave, and the zero solution. By nature of their method, the feedback controller must have access to the full solution at all times to compute the control. The authors also show that the number of actuators required to control the system (in their optimal control setup) is proportional to the number of unstable modes of the target solution. The authors conduct numerical experiments for certain choices of $\nu \geq 0.01$.

Zeng et al. (2022) combine a reduced-order model, trained on simulations of the KS system with random actuations, with an RL controller. The authors consider a domain of length $L = 22$ (or $\nu = 0.08156...$), which is characterised by oscillatory dynamics. The RL controller used is DDPG and it is trained using only the reduced-order model, which is comprised of an autoencoder and a neural ODE. The combination of reduced-order modeling with reinforcement learning holds great promise especially for more complex systems in active flow control, since it can potentially increase sample efficiency and eliminate demands for costly simulations. We remark that in order to learn the reduced-order model in (Zeng et al., 2022), access to the full solution is necessary. In a similar vein, Zeng & Graham (2021) apply a DDPG controller to the KS equation and show that using a symmetry reduction method improves performance. The authors consider the equation on a domain of length $L = 22$ (or $\nu = 0.08156...$, see above).

Özalp et al. (2023) demonstrate that a reduced order model for the KS equation in a fully chaotic regime ($\nu = 0.01$) can be learned using only partial observations and a physics-informed loss. The authors show that the time dynamics can be captured with an LSTM, which are typically simple and robust to train. We remark here that the use of LSTMs in this work differs somewhat from the way in which LSTMs are typically used in reinforcement learning: in order to predict the solution at time $t + 1$ from previous data, Özalp et al. (2023) use a time window of partial observations and map this sequence of observations through the LSTM, initialising the hidden state to zero. By contrast, in reinforcement learning, the hidden state of the LSTM is typically remembered across timesteps. Thus to compute an output at time $t + 1$, the hidden state from the previous time state is combined with one observation and mapped through the LSTM to update the hidden state, see Figure 2.

Xu & Zhang (2023) apply deep reinforcement learning to the control of the linearised KS equation, which models a convectively unstable flat-plate boundary layer flow in the vicinity of the boundary.

Finally, we point the reader to further literature on numerical studies of the KS equation, analysing bifurcation behaviour, long-term dynamics, chaotic attractor dimension and other quantities of interest, see for instance (Nicolaenko et al., 1985; Papageorgiou & Smyrlis, 1991; Linot & Graham, 2020). For the vast literature on analytical results for the KS equation, see for instance (Giacomelli & Otto, 2005; Otto, 2009; Foias et al., 1986).

