# OpenReview forum: "CHAROT: Robustly controlling chaotic PDEs with partial observations"
_ICLR.cc/2024/Workshop/AI4DiffEqtnsInSci — AI4DiffEqtnsInSci @ ICLR 2024 Poster_

### Official Review · Reviewer_BBj9 · 2024-02-26
**Authors propose CHAos-RObust-Transformers, an attention based memory architecture which augments actor-critic reinforcement learning algorithms for controlling chaotic PDEs using only partial observations.**

**Rating:** 9
**Confidence:** 4

**Review:**

Authors propose CHAos-RObust-Transformers, an attention based memory architecture which augments actor-critic reinforcement learning algorithms for controlling chaotic PDEs using only partial observations.

---

### Official Review · Reviewer_PW9E · 2024-02-27
**Review of CHAROT paper**

**Rating:** 7
**Confidence:** 3

**Review:**

This is a review of the manuscript, "CHAROT: Robustly controlling chaotic PDEs with partial observations," submitted to the ICLR 2024 Workshop on AI4DifferentialEquations In Science. The paper describes an attention-based memory architecture that can augment actor-critic reinforcement learning (RL) algorithms in the context of chaotic partial differential equations (PDEs). The authors compare the utility of three controllers for the purpose of forcing a Kuramoto–Sivashinsky (KM) system. Tested controllers include:
1. a simple (memory-less) multilayer perceptrion,
2. a controller with long short-term memory, and
3. a modified transformer, a.k.a. "CHAROT".
The CHAROT controller exhibits superior performance to the other two across a range of KS parameters, spanning the weakly- and fully-chaotic regimes. Information in the appendices describe the effects of sensor density, hyperparameter selection, architectural choices, etc.

The paper is well-written and the results seem convincing. As is often the case with such work, it is unclear whether the dimension of the three controllers has been held constant across the tests. I do not necessarily doubt the utility of the chosen architecture, but it seems unreasonable to compare controllers with highly divergent numbers of trainable parameters, i.e., because the size of the model typically has a first-order effect on performance. Moreover, I do not see how the proposed architecture has been tailored to deal with chaotic systems, in particular. Perhaps the authors can motivate their architectural choices with regards to features of chaotic systems.

Lastly, I question the relevance of such methods to real flow control problems. It seems to me that developing a reliable training protocol (including the requisite scale-resolving simulations) is far and away the dominant challenge in that context, and highly-turbulent systems resist selective control actions (turbulence has a way of "taking over"). It might be useful for the authors to sketch a practical application of the proposed control strategy, including the training data and expected sensor data.

---

### Meta-Review · Area_Chair_XMNT · 2024-03-01

**Recommendation:** Accept (Poster)

**Metareview:**

Authors propose CHAROT, an attention-based memory architecture to augment actor-critic reinforcement learning for controlling chaotic PDEs. CHAROT demonstrates superior performance over MLP and LSTM baselines. Based on reviews, I suggest adding justifications to the paper on architectural choices for chaotic systems, discussing real-world applicability, training data needs, and providing an application sketch to strengthen the work for the camera-ready version.

---

### Decision · Program_Chairs · 2024-03-02

Accept (Poster)